# Antarctic ozone hole modifies iodine geochemistry on the Antarctic Plateau

Andrea Spolaor [1,2✉], François Burgay [2,3], Rafael P. Fernandez[4], Clara Turetta [1,2], Carlos A. Cuevas [5], Kitae Kim[6], Douglas E. Kinnison[7], Jean-François Lamarque [7], Fabrizio de Blasi[1,2], Elena Barbaro[1,2], Juan Pablo Corella[5,8], Paul Vallelonga[9,10], Massimo Frezzotti [11], Carlo Barbante [1,2] & Alfonso Saiz-Lopez [5✉]

Polar stratospheric ozone has decreased since the 1970s due to anthropogenic emissions of chlorofluorocarbons and halons, resulting in the formation of an ozone hole over Antarctica. The effects of the ozone hole and the associated increase in incoming UV radiation on terrestrial and marine ecosystems are well established; however, the impact on geochemical cycles of ice photoactive elements, such as iodine, remains mostly unexplored. Here, we present the first iodine record from the inner Antarctic Plateau (Dome C) that covers approximately the last 212 years (1800-2012 CE). Our results show that the iodine concentration in ice remained constant during the pre-ozone hole period (1800-1974 CE) but has declined twofold since the onset of the ozone hole era (~1975 CE), closely tracking the total ozone evolution over Antarctica. Based on ice core observations, laboratory measurements and chemistry-climate model simulations, we propose that the iodine decrease since ~1975 is caused by enhanced iodine re-emission from snowpack due to the ozone hole-driven increase in UV radiation reaching the Antarctic Plateau. These findings suggest the potential for ice core iodine records from the inner Antarctic Plateau to be as an archive for past stratospheric ozone trends.

[1] Institute of Polar Sciences, CNR-ISP, Campus Scientifico Via Torino 155, Mestre, 30172 Venice, Italy. [2] Department of Environmental Sciences, Informatics and Statistics, University Ca'Foscari of Venice, via Torino, 155 - 30172 Venice-Mestre, Italy. [3] Paul Scherrer Institute, Laboratory of Environmental Chemistry, 5232 Villigen, PSI, Switzerland. [4] Institute for Interdisciplinary Science, National Research Council (ICB-CONICET), FCEN-UNCuyo, Mendoza 5501, Argentina. [5] Department of Atmospheric Chemistry and Climate, Institute of Physical Chemistry Rocasolano, CSIC, Madrid, Spain. [6] Korea Polar Research Institute, Incheon 21990, Korea. [7] National Center for Atmospheric Research, Boulder, CO, USA. [8] CIEMAT, Environmental Department, Av. Complutense 40, 28040, Madrid, Spain. [9] Physics of Ice, Climate and Earth, Niels Bohr Institute, University of Copenhagen, Tagensvej 16, Copenhagen N2200, Denmark. [10] UWA Oceans Institute, University of Western Australia, Crawley, WA 6009, Australia. [11] Department of Science, University of Roma Tre, Largo S. Leonardo Murialdo, 1, 00146 Roma, Italy. ✉email: andrea.spolaor@cnr.it; a.saiz@csic.es

The stratospheric ozone layer, which extends from the tropopause (≈17 km in the tropics and ≈11 km in the polar regions) to the upper stratosphere (≈35 km), completely absorbs the 220–290 nm component of the solar electromagnetic spectrum[1] and constitutes a natural shielding that reduces the amount of biologically harmful radiation reaching the Earth's surface within the 290–320 nm band[2]. In the mid-1970s, it was hypothesized that the massive emission of chlorofluorocarbons (CFCs) and halons, commonly used as propellants, solvents and refrigerants, could lead to a decrease in stratospheric ozone concentrations[3,4]. These compounds, when exposed to UV radiation, release halogen radicals in the stratosphere that lead to ozone destruction through catalytic cycles[5,6]. The effectiveness of CFCs and halons in depleting stratospheric ozone is maximized at the poles, where the Antarctic ozone hole was identified in the mid-1980s[7]. The effects of enhanced solar UV radiation, resulting from stratospheric ozone loss, on human health[8] and terrestrial[5,9] and marine[10] ecosystems have been well established. However, a limited number of investigations[11–13] have been performed to evaluate the impact of ozone hole formation and evolution on the geochemical cycle of photoactive species deposited on polar snow and ice.

In this work, we report the first iodine record from an ice core collected at Dome C (inner East Antarctic Plateau), covering the period of 1800–2012. Among the three available Antarctic iodine records[14,15], this is the only one that covers both the pre-ozone hole and the ozone hole periods and enables us to evaluate the influence of increasing incoming solar UV radiation on snowpack-atmosphere iodine exchange equilibrium during the ozone hole period (1975–2012). Combining field observations, laboratory measurements and chemistry-climate model simulations, we find that the iodine concentration in ice remained relatively stable during the pre-ozone hole period (1800–1974) but has gradually declined by a factor of 2 since the onset of the Antarctic ozone hole in the mid-1970s. We suggest that the enhanced incoming solar UV radiation reaching the Antarctic snowpack due to the formation of the Antarctic ozone hole has significantly amplified the ice-to-atmosphere iodine mass transfer, altering the natural geochemical cycle and mobilization of this essential element on the Antarctic Plateau.

## Results and discussion

**The Dome C iodine records**. In 1996, a French-Italian team established the first summer camp at Dome C, the inner Antarctic Plateau, and a new all-year facility, Concordia Station (3233 m a.s.l.; 75°05′59″S, 123°19′56″E, ~1200 km from the Southern Ocean), became winter-over operational during the 2005 season. A 13.72 m shallow ice core was drilled close to Concordia during 2012. This record covers ~212 years, from 1800 to 2012. The ice core chronology of the shallow core, which has an estimated ice age uncertainty of up to 5 years, is based on the annual snow accumulation and on the observed $nssSO_4^{2-}$ spikes from the most important past volcanic events[16,17] (see Supplementary Material S1, hereafter SM). Due to the very dry conditions, low precipitation (25–30 mm water equivalent per year, mm w.e. yr$^{-1}$) and very thin atmospheric boundary layer (surface pressure of ~650 hPa), Dome C is an ideal site to study snow photochemical processes and snow-atmosphere exchange of reactive elements[18–20].

The shallow ice core was sampled at 5 (±1) cm resolution, and for each sample, total iodine (I) and sodium (Na) were measured using an inductively coupled plasma sector field mass spectrometer (ICP-SFMS, see the "Methods" section). Both iodine and sodium are mainly emitted by seawater and sea ice within coastal regions and are transported inland[21,22]. As sodium does not experience photochemical degradation during transport or after deposition, it

is widely used as a conservative tracer[23]. To support the 13.72 m-deep shallow core, a 1.3 m-deep snow pit (Sp2013) and a 4 m-deep snow pit (Sp2017) were dug in December 2013 and December 2017, respectively (see S2 in the SM for further details).

The average iodine concentration for the entire ice core record was $0.056 \pm 0.037$ (1σ) ng g$^{-1}$. To detect a possible tipping point discontinuity within the iodine record, we used the Change-Point Analysis test (CPAt[24], see S3 in the SM). The results suggest that the iodine concentration record shifted in 1975 (p-value = 0.001) with a confidence interval of 90%. Based on the CPAt test, we identified two main periods for the iodine record, one from 1800 to 1974 and the other from 1975 to 2012. The average iodine concentration for the period of 1800–1974 remains constant at $0.060 \pm 0.039$ ng g$^{-1}$, decreasing by almost half to an average concentration of $0.032 \pm 0.015$ ng g$^{-1}$ for the period of 1975–2011 (Fig. 1).

**UV radiation reaching the Antarctic Plateau**. To evaluate the hypothesis that the increased UV radiation reaching the Antarctic Plateau due to ozone hole formation has altered iodine concentrations within the ice core, we computed the total ozone column (TOC) and ozone hole evolution during the 1950–2010 period using the CAM-Chem chemistry-climate model[25,26] (see "Methods"). Remarkably, the modelled TOC trend above Dome C (TOC$^{DC}$) matches the iodine record both before and after the observed tipping point discontinuity (Fig. 1c), which coincides with the initial stages of ozone hole formation in ~1975[5]. Furthermore, the modelled mean actinic flux at 300 nm (AF$^{300}$, Fig. 1a) within the closest model grid point to Dome C (74.84° S; 122.5°E; grid resolution 1.9° × 2.5°, see "Methods") completely mirrors the iodine record and total ozone trend (Fig. 1c), highlighting the large radiative changes affecting the inner Antarctic surface due to ozone hole formation.

We computed the mean modelled TOC$^{DC}$ and AF$^{300}$ during the sunlit period of each year (i.e., from 1 September to 28 February) before and after the iodine tipping point and considered these values to be representative of the pre-ozone hole (1950–1974) and ozone hole (1975–2011) periods. Within these periods, the modelled TOC$^{DC}$ decreased from $334 \pm 16$ DU (1950–1974) to $266 \pm 41$ DU (1975–2017). Consequently, this stratospheric ozone reduction drove an increase in AF$^{300}$ at Dome C from $1.05 \times 10^{11}$ to $4.36 \times 10^{11}$ quanta cm$^{-2}$ s$^{-1}$ nm$^{-1}$. Note that the mean ozone reduction and the actinic flux enhancement reaching the inner Antarctic surface strongly depend on the specific wavelength, location and period (i.e., early spring or mid-summer) considered. A sensitivity analysis of the spatiotemporal variability in these quantities is provided in section S6 in the SM.

During the ozone hole period, the correlation between [I] and sunlit TOC$^{DC}$ has a positive and significant Pearson coefficient, whereas no significant correlation was found for the pre-ozone hole period (pre-1975: $r = -0.1$, p-value = 0.65; post-1975: $r = 0.398$, p-value = 0.015; see S3.3 in the SM). Similarly, [I] and UV forcing, expressed as AF$^{300}$, were uncorrelated before 1975, while they were anti-correlated afterwards, indicating a direct effect induced by the stronger UV radiation reaching the Antarctic Plateau (pre-1975: $r = -0.018$, p-value = 0.93; post-1975: $r = -0.401$, p-value = 0.015; S3.4 in the SM). These significant correlations suggest a direct linkage between stratospheric ozone and UV radiation reaching the inner Antarctic Plateau surface, which in turn would alter the postdepositional processes controlling the preservation of iodine within the snowpack, as described below.

**Role of ozone hole in the iodine geochemical cycle**. To evaluate the physicochemical processes behind the observed negative

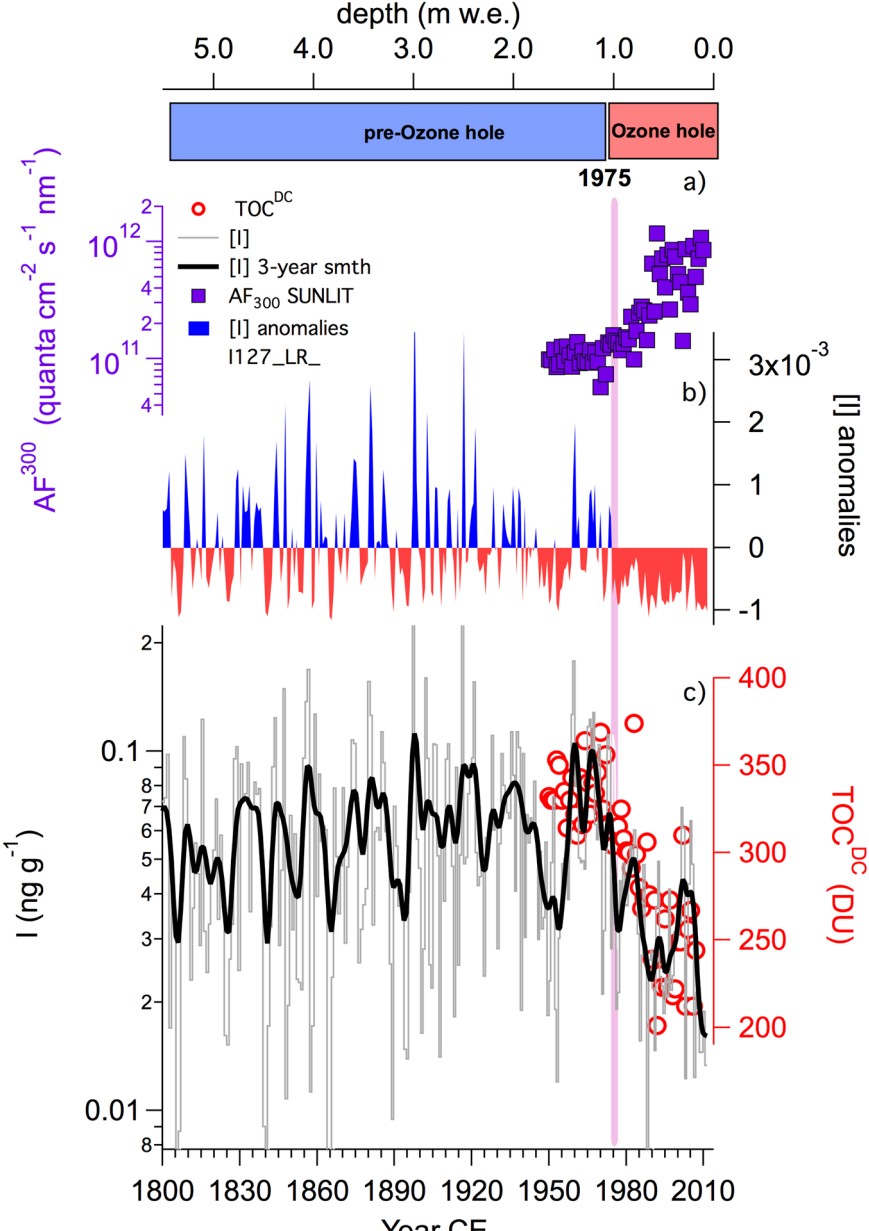

**Fig. 1 Evolution of iodine concentration and stratospheric ozone over Dome C.** Iodine concentration (grey line, raw data; bold black line, 10 year smoothing) and [I] anomalies, expressed as ng g$^{-1}$ (blue, positive; red, negative; solid lines with respect to the mean iodine concentration for the period of 1800–2012), were compared with the mean total ozone column above Dome C during the sunlit (1 September to 28 February) period of each year (red dots, TOC$^{DC}$ in Dobson units). The top panel shows the modelled mean sunlit Actinic Flux at 300 nm (purple square, AF$^{300}$, quanta cm$^{-2}$ s$^{-1}$ nm$^{-1}$) at the Antarctic surface within the closest model grid point to Dome C. The decrease in iodine concentration coincides with the decrease in TOC$^{DC}$ and mirrors the increase in modelled AF$^{300}$. The pink vertical bar shows the tipping point determined for [I] (1975 CE), which is considered to be the starting date for the ozone hole period.

correlation between UV surface radiation and ice core iodine, we turn to condensed phase iodine photochemistry. Within liquid aerosols and ice crystals, inorganic iodine is mostly in ionic form as iodide (I$^-_{(aq)}$) and iodate (IO$_3^-_{(aq)}$)[27]. Although iodate is supposed to be more stable than iodide, several studies highlighted that the actual I$^-_{(aq)}$/IO$_3^-_{(aq)}$ ratio depends on the sampling location, with I$^-_{(aq)}$ being the most abundant species in polar environments[28]. Indeed, investigations performed on the Talos Dome Antarctic ice core (www.taldice.org 72°49'S, 159°11'E, 2315 m a.s.l., accumulation of 80 mm w.e. yr$^{-1}$) highlighted that I$^-_{(aq)}$ is more stable than IO$_3^-_{(aq)}$[15]. Both iodine species showed strong photoactivity in snow matrices since they

are enriched during the polar night and depleted during summer[29–31]. Under laboratory and simulated Antarctic radiation conditions, it was observed that iodate UV absorption and its subsequent photoreduction (IO$_3^-_{(aq)}$ + hv → IO$_{(g)}$ + O$_2^-_{(aq)}$) were enhanced in the 275–400 nm range, with a maximum peak at 295 nm[32]. Similar conclusions were also reported for the photooxidation of iodide into I$_{2(g)}$ in real snow samples spiked with known amounts of iodide and exposed to Antarctic sunlight (O$_{2(aq)}$ + 4H$^+_{(aq)}$ + 6I$^-_{(aq)}$ → 2H$_2$O + 2 I$_3^-_{(aq)}$; I$_3^-_{(aq)}$ ↔ I$_{2(g)}$ + I$^-_{(aq)}$)[33]. Within this photooxidation mechanism that leads to the release of I$_{2(g)}$ to the atmosphere, a charge-transfer complex with oxygen (I$^-$-O$_{2(aq)}$) is formed. This complex has a

local absorption maximum between 280 and 330 nm, i.e., encompassing the same spectral range where the largest changes in incoming UV radiation were observed between the pre- and post-ozone hole periods (Fig. 2a).

Given that iodide abundance in polar ice is higher than iodate, we conducted a timeline simulation of iodine photoactivation within ice and snowpack (J-iodine, Fig. 2b) by computing the wavelength integral of the measured iodide absorption spectra multiplied by the modelled AF reaching the inner Antarctic surface during the 1950–2010 period (see "Methods"). We find that the normalized J-iodine trend shows consistent agreement with the iodine decrease observed for the Dome C ice record. This finding suggests the occurrence of UV-driven re-emission and mobilization of snow-trapped iodide to the gas phase, following the photooxidation of iodine in ice and subsequent release of $I_{2(g)}$ to the atmosphere[34]. Indeed, J-iodine remained relatively constant during the pre-1975 period, then increased ~1.5-fold for the mean ozone hole era and ~2.0-fold for the 2006–2012 period. This change is indicative of a continuous change in the efficiency of re-emission processes of ice-trapped iodine that altered the steady state equilibrium, assuming that the iodine deposition velocity remains unaltered (see next section). Due to the pronounced changes in AF intensity reaching the inner Antarctic surface above and below 300 nm during different

seasons, the post-1975 J-iodine enhancement computed here is very sensitive to the upper bandwidth limit and to the month used to perform the wavelength integration. Indeed, a sensitivity analysis indicates that up to fourfold more efficient iodide photodissociation efficiency is obtained when the narrower 280–307 nm AF band is considered, while an equivalent ~5-fold J-iodine enhancement is reached if only the October mean evolution is used (see Supplementary Figs. 5 and 6). Therefore, regardless of the exact wavelength interval and seasonal period considered, our combined experimental and modelling results indicate that the increase in UV solar radiation observed since the onset of the ozone hole has promoted the release of active iodine from the snowpack to the atmosphere. This validation corroborates our hypothesis that stronger UV radiation led to the enhanced re-emission of iodine deposited on polar ice, thereby explaining the overall decrease in iodine concentration at Dome C during the 1975–2012 period.

**Possible competitive processes (CPs).** To explain the observed shift in the iodine trend at Dome C, we explored different alternative hypotheses that involve possible competitive processes (CPs), such as (a) changes in site snow accumulation, (b) dependence on snow pack physical characteristics, (c) snow pack postdepositional processes and spatial variability, (d) changes in

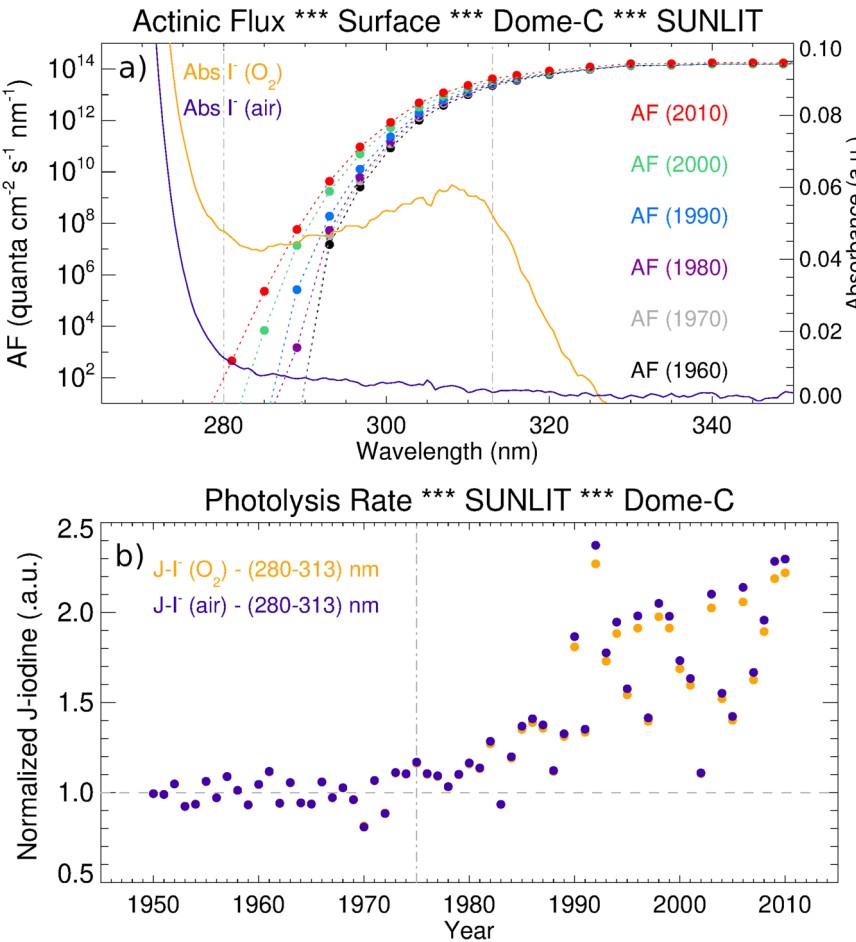

**Fig. 2 Iodine photochemical enhancement during the ozone hole period.** Modelled actinic flux (AF) and iodine photoactivation efficiency (J-iodine) at the snow surface in Dome C. **a** Wavelength-dependent AF reaching the model surface every decade since the 1960s (colour dots, left Y-axis). The right Y-axis shows the absorbance spectra of a frozen iodide solution in equilibrium with oxygen (I⁻(O₂)) and air (I⁻(air)) environments. **b** The normalized J-iodine of snow-trapped iodine between 1950 and 2010 relative to the mean J-iodine for the pre-1975 period (dashed-grey horizontal line in (**b**)). J-iodine wavelength integration was performed within the 280–313 nm band (see dashed-grey vertical lines in (**a**)) and continuously increased after 1975 due to the ozone hole-driven enhancement in UV radiation during the sunlit period of each year (i.e., from 1 September to 28 February).

the transport mechanism from the coast to the site, and (e) changes in the strength of coastal source emissions. In the SM (S7), we also discuss the possible effects of sample storage on iodine loss from firn samples. All these alternative hypotheses were refuted, as they failed to provide a consistent explanation that connects theory with observations, as discussed below. Additional iodine ice core records exist in both the Arctic and Antarctic and are discussed in the SM (S4 and Supplementary Fig. 1); however, these records do not provide additional information for explaining the Dome C iodine trend.

**CP(a) – Change in snow accumulation**. The inner Antarctic Plateau is characterized by low accumulation rates and cold temperatures that can affect snow metamorphism. Due to the very low annual snow deposition[35], it is possible to estimate snow accumulation at Dome C only at pluriannual scale using snow/firn core analysis. However, the average snow accumulation rate computed during different studies is very similar: for the period of 1955–1998, the rate was $26 \pm 1.3$ mm w.e. yr$^{-1}$; for the period of 1816–1998, it was $25 \pm 1.3$ mm w.e. yr$^{-1}$ [36]; and for the 2006–2013 period[35], the rate was 27 mm w.e. yr$^{-1}$. This result leads to the conclusion that the snow accumulation rate at this location has not changed significantly over the last 200 years and that in the worst scenario, it might have affected the iodine signal by no more than 10%. Note that the spatial variability in accumulation might lead to interannual variability but will not affect the recorded trend (Supplementary Fig. 4, lower panel and Supplementary Table 2). For this reason, we assume a constant snow accumulation rate over the entire period covered by the ice core record, and we discard the possibility that snow accumulation changes can explain the large observed reduction in iodine concentration since 1975.

We also investigated the possible role of the precipitation seasonality at Dome C. Combining a Lagrangian trajectory model output and the European Re-Analysis ERA40 data (from 1980 to 2011), it is reported that ~35–40% of the annual precipitation at Dome C occurred during the summer season (December to February) with the remaining 60–65% being homogenously distributed among the remaining months[37]. We found that the seasonality of snow precipitation is negligible due to the wind redistribution of snow accumulation at the surface[38]. UV can penetrate the snowpack for the first 20–30 cm, which corresponds to 2–3 years of burial snow due to the low snow accumulation rate at Dome C[38]. Independently of the seasonal precipitation pattern, all the iodine present in the annual snow layers would experience a similar photochemical activation, thereby greatly minimizing the hypothetical role of a seasonal precipitation pattern change. However, the potential impact of systematic changes in precipitation seasonality over several years remains an uncertainty for the interpretation of future iodine ice core records and might contribute to the observed spatial variability.

**CP(b) – Role of snow pack physical characteristics**. The snow density can affect the chemical signal both by modifying the extent of light penetration and by changing the gas permeability within the snow pack. The average snow density at Dome C increases with depth, making the snow layers progressively less permeable to gas movement until the close-off depth is reached (≈80 m depth). The gaseous species can move within the snow/ice grains until reaching this depth, but their escape into the atmosphere is potentially facilitated by the higher porosity of the surface layers (i.e., with a lower density). However, the decline in iodine concentration was observed only in the topmost 3.5 m (1.2 m w.e.) of the snow pack (Fig. 1c), and the direct comparison with the Sp2017 iodine and density profiles does not endorse this

hypothesis (see Supplementary Fig. 3). Indeed, the Sp2017 density profile presents a typical shallow density increase in the topmost 3.5 m (from 0.32 to 0.38 kg L$^{-1}$), which cannot justify the observed iodine behaviour.

Another process that can affect the stabilization of iodine species in snow is grain size. The grain size increases with depth[40], reducing the specific surface where gas-phase iodine can be bound and/or absorbed. If a relationship between the grain size and iodine concentration existed, we might expect an enhancement of iodine concentration in the topmost part of the record, but we observe the opposite. Snow grain size was measured in Sp2017, and similar to snow density, it cannot explain the iodine trend (Supplementary Fig. 3).

Finally, snow metamorphism is mainly driven by the temperature gradient prevailing in the first metres of snow between the polar summers and winters and occurs on timescales on the order of weeks or months. Continuous snow temperature measurements are routinely performed at Dome C down to 10 m depth[41]. The largest difference in temperature during the year is measured in the uppermost 2 m, particularly at the surface where the $\Delta T$ can reach 44 °C. Note that the largest measured temperature gradient at 2.5 m depth is 8 °C (Supplementary Table 3), which slows snow metamorphosis and possible related processes. In our shallow core, the iodine shift is detected at a depth of 3.5 m. Moreover, when considering iodine postdepositional processes within the snow pack, we need to consider that iodine photochemical activation is fast and can cause a loss of up to 90% of the deposited iodine within a day[31,42]. The radiative transfer within the snowpack will be further discussed in the next paragraph, but it suggests that the snow metamorphism, the physical characteristics of the snowpack and its sintering process play a negligible role in describing the observed iodine trend.

**CP(c) – Inner snow pack postdepositional processes and UV light penetration**. We also investigated the effects of UV penetration into snowpack. UV light can penetrate the snowpack for the first 20–30 cm[39], corresponding to 2–3 years of burial due to the low snow accumulation rate at Dome C. UV light penetration would have caused the re-emission of iodide as gaseous I$_2$ into the atmosphere for ~2–3 years. However, if this case were true for iodine, we would expect a similar iodine concentration along the entire core or, alternatively, iodine accumulation would occur in the surface layers due to the migration of gaseous iodine from deeper sections of the core. Evidence that UV penetration in snowpack does not play a significant role also comes from the availability of additional iodine records (i.e., Sp2013 and Sp2017) that showed similar trends and concentrations over the last 40 years (Supplementary Table 2 and Supplementary Fig. 4). Furthermore, our iodine analyses from the Law Dome (LD) ice core, which has an average annual snow accumulation of ≈740 mm w.e. yr$^{-1}$ (i.e., ~30 times Dome C), showed well-preserved iodine peaks corresponding to the winter periods (polar night)[14,29]. If light penetration determined re-emission from the snowpack, we would expect a smoother signal in the record compared with the observed sharp peaks. On these grounds, we argue that the iodine trend is influenced by the enhancement of UV radiation and the increase in photoreactivity on the surface snow instead of depending on postdepositional processes occurring within the compacted deep ice. This finding is also supported by the different trend of iodine, which decreases from the surface to 3.5 m (1975 CE), in comparison with other photochemical species analysed in Sp2017 (Supplementary Fig. 4).

**CP(d) - Changes in the transport mechanism from the coast to the site**. To understand whether changes in the transport

mechanisms from the coast to Dome C played a role in explaining the iodine behaviour, we evaluated how sodium concentration has changed over the last two centuries. Overall, the sodium average concentration for the entire record was $41 \pm 19$ ($1\sigma$) ng g$^{-1}$; however, a slight increase was recorded from 1975 to 2012 ($50 \pm 23$ ng g$^{-1}$) compared with the period spanning 1800 to 1975 ($39 \pm 14$ ng g$^{-1}$). Note that the ice core record had some gaps from 1989 to 1997 that were filled with Na concentration data retrieved from Sp2013 and Sp2017 (see S3.5 in the SM), producing a merged sodium ($Na_M$) and iodide ($I_M$) record for direct comparison.

Comparing the $Na_M$ and $I_M$ profiles, we observe that they have opposite trends during the last few decades (Fig. 3). The increase in sodium concentration recorded since the mid-1970s suggests enhanced transport from the coast to Dome C. This finding is consistent with model simulations and observations that report a poleward shift in the mid-latitude jet stream[43,44] and with the enhanced cyclonic activity driven by the phase change in the Interdecadal Pacific Oscillation (IPO) as well as strengthening of westerlies due to global warming[45]. However, the enhanced

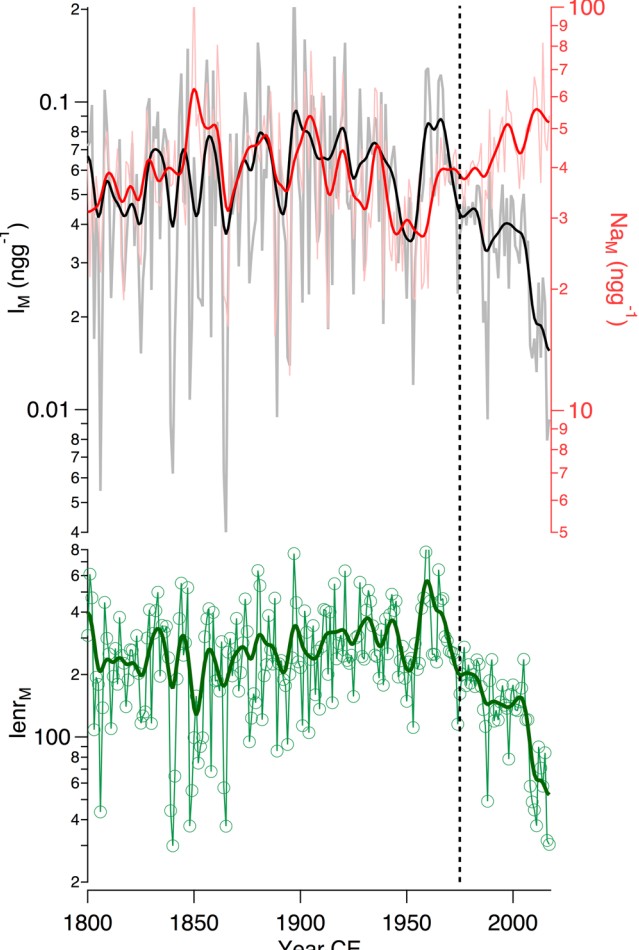

**Fig. 3 Iodine and sodium merged record from Dome C.** The upper panel shows the merged 1-year smooth records (see S3.5 in the SM for details) using snow pits and ice core data (iodine merged in black— $I_M$ - and sodium merged in red— $Na_M$). Light grey and red lines show the raw data, and red and bold curves show the 5-year smoothing. The lower panel shows the iodine enrichment ($I_{enr} = I/(Na*0.00000568$, where 0.00000568 is the I/Na mass ratio in seawater), where the solid green line shows the 5-year smoothing. The dashed vertical line indicates the tipping point discontinuity at 1975 CE.

poleward transport is not mirrored by the iodine record, which, conversely, has decreased since 1975. The decoupling between $Na_M$ and $I_M$ is corroborated by the Pearson correlation coefficients calculated for the periods of 1800–1974 and 1975–2017 (see S3.5 in the SM). A significant statistical correlation existed between iodine and sodium during the pre-ozone hole period (1800–1974; $r = 0.44$; $p$-value $< 0.0001$) but not for the ozone hole period ($r = -0.1$; $p$-value $= 0.54$). Thus, the statistical tests support that between 1800 and 1974, iodine and sodium were transported and deposited following similar mechanisms. Once deposited on the snowpack, iodine experienced postdepositional processes whose efficiency remained relatively constant throughout the entire pre-ozone hole period investigated in this study (Figs. 1 and 3). The decoupling observed since 1975 is inconsistent with an iodine decrease driven by changes in atmospheric transport. All of these previous studies provide additional evidence that the different behaviours of Na and I from 1975 to 2017 at Dome C were associated with processes that interfered with iodine deposition and preservation in the snowpack rather than with changes in the snow accumulation rate, coastal source strength or transport efficiency. In fact, enhanced tropospheric transport from the coast to the Antarctic inland[45] would have caused an increase in both Na and I concentrations.

**CP(e) - Possible changes in the strength of coastal source emissions**. Finally, we explored the possible changes in the strength of iodine emissions from coastal areas. However, this hypothesis is not supported by models or observations. Note that the most relevant iodine source on a global scale is the ocean, providing ~2.3 Tg(I) yr$^{-1}$ of organic (mostly $CH_3I$) and inorganic ($I_2$ and HOI) iodine[46,47]. Global iodine emissions show strong latitudinal variability, with stronger emissions in the tropical and mid-latitude oceans and significantly decreasing emissions towards the poles[48]. In polar regions, iodine sources are mainly connected to first-year sea ice[49,50], with both macro- and microalgae living under sea ice[51] and abiotic activation of iodine on sea ice and snow surfaces[22,52] playing a role. Both the biological and abiotic iodine emissions in the polar regions depend on the thickness and extent of seasonal sea ice, being more prominent for thinner sea ice and large extensions of first-year sea ice. A trend of increasing Antarctic seasonal sea ice extent has been observed for the last decades up to 2014, followed by a decline from 2015 to 2017[53]. Thus, it would be expected that in any case, coastal sea ice emissions of iodine increased during the studied period, contrary to what we observed in the ice core record.

This study suggests that the human-induced stratospheric ozone hole has altered the natural geochemical cycle of iodine. Combining field measurements (ice core, snow pit, surface snow observations) with laboratory (iodine absorption spectra, UV-driven photooxidative mechanisms) and modelling (ozone hole evolution and associated actinic flux changes) results, we conclude that the enhanced incident UV radiation due to the ozone hole has caused a continuous decline in iodine concentration in ice in inner Antarctica since ~1975. This recent phenomenon has modified the previously steady iodine equilibrium between snowpack and the atmosphere in the interior of Antarctica by increasing re-emissions from snowpack. The resulting increase in ice-to-atmosphere iodine mass transfer has relevant implications for polar tropospheric chemistry and for the Earth's radiative budget since iodine catalytic cycles play a crucial role in the destruction of tropospheric ozone[26,27,54,55] and can also act as cloud condensation nuclei (CCN)[56–58]. Furthermore, surface re-emission of iodine from central Antarctica and subsequent redistribution by atmospheric transport could

potentially be a source of the widespread distribution of iodine observed by satellite measurements over the entire Antarctic continent[59]. Finally, given the direct link observed between ice core iodine at Dome C and stratospheric ozone hole evolution, we suggest that the ice core iodine present on the Antarctic Plateau may potentially serve as an archive for past stratospheric ozone changes at centennial to millennial time scales.

## Methods

**Shallow core processing and analysis.** The shallow core was cut and sampled at $5 \pm 1$ cm resolution using a ceramic knife and rinsed with ultrapure water (UPW) after each use. Only the central part of the core was used for the analysis, and the outer 2 cm was removed by scraping with a ceramic knife. The core samples were processed at the Institute of Polar Sciences of the National Research Council (ISP-CNR) in a class 1000 inorganic clean room under a class 100 laminar-flow bench. Iodine and sodium analyses in the snow pit and ice core samples were conducted on melted and non-acidified samples. Total sodium (Na) and iodine (I) concentrations were determined by ICP-SFMS following Spolaor et al.[60]. Each analytical run started and ended with a UPW cleaning session of 3 min to ensure a stable background level throughout the analysis. The external standards that were used to calibrate the analytes were prepared by diluting a 1000 ppm stock IC (ion chromatography) standard solution (TraceCERT® purity grade, Sigma-Aldrich, MO, USA). The standard concentrations ranged between 1 and 200 ng g$^{-1}$ for sodium and 0.005 and 0.200 ng g$^{-1}$ for iodine. The relative standard deviation (RSD %) was low for all analytes, ranging from 3 to 4% for sodium and 2 to 5% for iodine. The instrumental limit of detection (LoD), calculated as three times the standard deviation of the blank ($n = 10$), was 1 ng g$^{-1}$ for Na and 0.003 ng g$^{-1}$ for I. We evaluated the stability of iodine by repeating the analysis of a selected sample multiple times. We did not observe any significant iodine loss during the analytical run ($\approx$12 h).

**CAM-Chem chemistry-climate model setup: Halogen chemistry and radiative transfer.** The stratospheric ozone hole evolution, as well as the radiative transfer of ultraviolet (UV) actinic flux reaching the Antarctic surface, was computed using the global 3-D chemistry-climate model CAM-Chem (Community Atmospheric Model with chemistry, version 4)[61], which is the atmospheric component within the CESM framework (Community Earth System Model)[62]. The model setup follows the CCMI-REFC1 recommendation for 1950–2010 simulations (i.e., we forced the model with prescribed sea surface temperatures and sea ice distributions)[63] but incorporates an explicit treatment of very-short lived (VSL) chlorine, bromine and iodine sources and chemistry[46,47,64–66]. The model configuration used here is identical to the one used previously to compare CAM-Chem results with Arctic iodine ice records[26], although here, we extracted monthly output with the wavelength-dependent actinic flux at all atmospheric layers and the surface. Additional details of the CAM-Chem configuration and validation are provided in S5 of the SM.

CAM-Chem shows a good ability to reproduce the size and depth of ozone hole evolution within chemistry-climate simulations[25] and presents excellent agreement with satellite ozone observations during the modelled period (Supplementary Fig. 2). The total ozone column polar cap (TOC$^{SP}$) used for the model validation was computed as the cosine-weighted mean within the 63°S–90°S latitudinal band during October. The model mean output during the austral sunlit period (i.e., from 1 September to 28 February of next year) was computed to perform the correlations with the ice core record. The complete sunlit period (spring + summer) was selected because even when the largest changes in TOC and AF$^{300}$ associated with the ozone hole are observed during spring, the overall intensity of UV radiation reaching the Antarctic surface maximizes during the summer because of the higher solar zenith angle (SZA). The 300 nm wavelength bin used for the actinic flux computation at the model surface (AF$^{300}$) has an ~3.5 nm width and is centred at 300.5 nm. AF$^{300}$ values were extracted at the grid point closest to Dome C on the Antarctic Plateau (74.84° S; 122.5°E; grid resolution 1.9° × 2.5°; model mean altitude of 3300 m a.s.l.; see S5 in the SM).

The iodide photolysis rate constant within the snowpack (J-iodine) was determined offline following Eq. (1)

$$J - \text{iodine}(\text{yr}) = \int \sigma^{\text{Abs}}(\lambda) \times \text{AF}^{\text{surf}}(\lambda, \text{yr}) \times \Phi(\lambda)\, d\lambda \quad (1)$$

where $\sigma^{\text{Abs}}(\lambda)$ is the absorption spectrum of iodide as a function of wavelength ($\lambda$) measured in iodine-containing frozen solutions[33]; AF$^{\text{surf}}(\lambda, \text{yr})$ is the wavelength-dependent surface CAM-Chem actinic flux reaching Dome C for each year (yr); and $\Phi(\lambda)$ is the quantum yield for the initial UV absorption step leading to the formation of triiodide in the frozen solution, assumed to be unity ($\Phi(\lambda) = 1$). The wavelength integration used to compute annual J-iodine was performed for the 280–313 nm bandwidth; below 280 nm, the transmittance of the Pyrex filter used for measuring $\sigma^{\text{Abs}}(\lambda)$ was considerably reduced, while above 313 nm, the iodide absorbance of iodide and the I$^-$-O$_{2(\text{aq})}$ complex rapidly dropped to zero[33]. The resolutions of the CAM-Chem wavelength bins used to compute J-iodine are

provided in the SM. Due to the rapid increase in AF$^{\text{surf}}$ intensity within the 290–330 nm range, as well as on the seasonal changes in the net solar radiation reaching the polar regions between spring, summer and fall, the absolute J-iodine computation is very sensitive to the upper bandwidth limit used for wavelength integration and to the monthly period of time considered for each year. A sensitivity analysis is provided in section S6 of the SM (Supplementary Figs. 5 and 6).

## Data availability
The datasets generated during and/or analysed during the current study are available from the corresponding author on reasonable request.

## Code availability
The software code for the CESM model is available from http://www.cesm.ucar.edu/models/.

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

## Acknowledgements

This project received funding from the European Union's Horizon 2020 Research and Innovation program under grant agreement no. 689443 via project iCUPE (Integrative and Comprehensive Understanding on Polar Environments), part of the European Commission (ERA-PLANET) and by the "Programma Nazionale per la Ricerca in Antartide" (PNRA, project number PNRA16_00295). This study also received funding from the European Research Council Executive Agency under the European Union's Horizon 2020 Research and Innovation programme (Project 'ERC-2016-COG 726349 CLIMAHAL'). NCAR is sponsored by the National Science Foundation under Grant Number 1852977. R.P.F. would like to express thanks for the financial support from CONICET-UNCuyo (SIIP-06/M111) and ANPCyT (PICT 2015-0714). This research was also supported by the Korea Polar Research Institute (KOPRI) project (PE20030) and by the Grant to Department of Science, Roma Tre University (MIUR-Italy Dipartimenti di Eccellenza).

## Author contributions

A.S. and A.S.-L. devised the research; A.S., F.B., R.P.F. and A.S.-L. wrote the manuscript; F.B., C.T., E.B. and A.S. prepared the ice core and snow pit samples and ran the chemical analysis; F.D.B ran the statistical analysis; R.P.F., C.A.C. and A.S.-L. ran the chemistry-climate model; K.K., D.E. K. and J.-F.L. provided the laboratory data and helped with the model configuration; and C.B., P.V., J-P.C., M.F. and C.A. helped with the data interpretation and manuscript writing.

## Competing interests

The authors declare no competing interests.
