## [Peer Review File · Nature Communications]

Antarctic ozone hole modifies iodine geochemistry on the Antarctic plateauReviewers' Comments:

Reviewer #1:

Remarks to the Author:

In this study, an iodine decrease in a shallow ice core from Dome C is used as a proxy for ozone hole depletion in the latter half of the 20th century. Laboratory measurements, atmospheric modelling, and comparison to the ice core sodium record are used to validate this interpretation. The authors provide a clear and concise explanation of their findings. This study should be published because it is the first time, to my knowledge, that the reemission of iodine from surface snow has been convincingly used as a proxy for increased UV-radiation reaching the Antarctic ice sheet surface. According to the authors, this is also only the third Antarctic iodine record to be published, and these other two records were incapable of reconstructing ozone hole depletion. The science described in this manuscript is sound. This interpretation of iodine could be very useful for future ice core studies. I just have a few minor comments. My only major comment is that the writing in both the main text and the supplement should be somewhat improved prior to publication. Some examples of possible grammatical changes are listed at the end of this review.

Major comments

The authors explain that UV-radiation reaching the snow surface does not affect the sodium record. Therefore, the sodium record is a good component to compare to the iodine record because otherwise, the temporal variability two records should (and do) correlate quite closely. Are there other halogens that are affected by the ozone hole that could or have been measured in ice cores?

In the supplement, the authors emphasize that this is only the third Antarctic iodine record to be published. It is also the only one that can be used to reconstruct ozone hole depletion. This is an excellent point. Can it be emphasized in the introduction somehow?

Minor comments

Change "well-studied" to "well-established" in the introduction and abstract

Line 69: remove "at present"

Lines 84-85: "due to its very dry.." and "its very thin..."

Line 92: "degradation during transport or after deposition"

Lines 93-95: "To support to the 13.72 m-deep shallow core, a 1.3 m-deep snow pit (Sp2013) and a 4 m-deep (Sp2017) snow pit were dug..."

Line 140: "mostly in ionic form as iodide..."

Lines 97-98: "we used the Change-Point Analysis test"

Line 199-200: change to "it was" in all cases

Line 201: change to "worst"

Line 203: change to "but will not affect the"

Line 216-220: confusing, please clarify

Line 223: "If a relationship between the grain size and iodine concentration existed,"

Line 225: "but we observe"

Line 228: "on timescales"

Line 229: Continuous measurements of what?

Line 231: "at the surface"

Line 233: "is detected"

Line 242: "2-3 years of burial"

Line 248: "Evidence that"

Line 249: "also comes from"

Line 259: "to 3.5 m"

Line 272: "last few decades"

Lines 286-287: "All of these previous studies"

Lines 298-299: "mid-latitude oceans, and significantly decreasing emissions towards"

Lines 299-300: please clarify: "In the polar regions, iodine sources mainly and first year sea ice"

Line 311: "has altered"

Lines 236-239: "Radiative transfer within the snowpack will be further discussed in Section 4.2.3, but its rapidity suggests that snow metamorphism, its physical characteristics and sintering process of the snow pack play a negligible role in describing the observed iodine trend." What is "its" referring to here? Please clarify this sentence.

Reviewer #2:

Remarks to the Author:

In the manuscript, the authors describe results of iodine determinations in ice cores from the Dome C station, which is located on the polar plateau of East Antarctica at about 3000 m altitude. The core discussed covers a little more than 200 years. The main result of the measurements is the observed decrease in total iodine concentrations in the ice since about the mid-1970s, which the authors link to the occurrence of the Antarctic ozone hole and the associated increase in UV radiation in this region. The authors' central hypothesis is that the increased UV intensity drives photochemical processes in the upper ice layers that accelerate the release of the halogen into the gas phase, thereby interfering with the geochemical cycles of the trace element. The hypothesis is supported by the discussion presented in various chapters of parameters such as UV intensity reaching the ground, photochemistry of various iodine compounds in the condensed phase, change in snow accumulation or physical characteristics of snow, transport mechanisms, etc.

Without question, iodine is an interesting element which not only shows an active redox chemistry, but the involved reduced and oxidized species also show different volatility and thus can migrate between different spheres (biosphere, pedosphere, hydrosphere, atmosphere, cryosphere), i.e. the element has the prerequisites to possess an active biogeochemical cycle. In fact, iodine has already been proposed as a paleoredox indicator. In my opinion, the paper is an excellent scientific work, in which, based on a comprehensive data set using various tools (trace analytical methods, chemical and transport models), an interesting relationship is demonstrated, in this case on the complex influence of photochemically-induced processes on important biogeochemical trace substance cycles. Therefore, I would recommend publishing the manuscript after minor changes.

Page 9: line 285: "The decoupling observed since 1975 is inconsistent with an iodine decrease driven by changes in atmospheric transport."!?

Page 10, line 299: The wording needs to be changed here, the sentence makes no sense as it is. Perhaps: "In the polar regions, iodine sources are mainly connected to first year sea ice, with both macroalgae and microalgae living under the sea ice and abiotic activation of iodine on sea ice and snow surfaces playing a role"!?

Page 10, line 320: While tropospheric ozone depletion is well supported with citations (which is only of regional importance), the influence of iodine on atmospheric particle formation (which may also have supraregional importance) is only very sparsely supported and cited. Here one could still improve.

Reviewer #3:

Remarks to the Author:

This study reports the first iodine firn core record from the East Antarctic Ice Sheet (EAIS) that covers about the last 200 years. The results indicate that the firn core iodine concentrations remained fairly constant during the pre-ozone hole period (1800-1974 CE) but have declined twofold since the onset of the ozone hole era (~1975 CE), closely tracking the stratospheric ozone above Antarctica. This is an interesting study highlighting the potential impact on iodine chemistry above Antarctica due to enhanced surface-UV from the modern ozone hole. The chosen statistical methods and modelling approach are sound and yield convincing results. My concerns are related to the presented firn core concentrations of [I].

It is well known that post-depositional exchange driven by photochemical and physical processes alters concentrations of a range of snow impurities before archival in firn and ice, and this has been the subject of numerous studies (e.g. Grannas et al., 2007). A quantitative understanding of the air-snow exchange is required before attempting to interpret the ice core record in terms of past atmospheric change.

Not doing so can lead to premature interpretation, as illustrated by the case of nitrate, where the modern increase in nitrate seen in EAIS pit profiles was linked to polar stratospheric clouds and the ozone hole, whereas later research showed that nitrate spikes in the low-accumulation regions reflect post-depositional air-snow recycling rather than change in an atmospheric source (e.g. Savarino et al., 2007).

Total iodine in surface snow is in so far complicated because it is subject to both photochemical loss as well as physical volatilization. In particular Legrand et al. (2018) found that due to the high volatility of some of the iodine species such as hypoiodous acid (HOI), iodine concentrations in ice strongly depend on the annual snowfall rate, particularly at sites with low snowfall rates.

Thus, I am concerned that the presented results are affected by such post-depositional loss in two ways

i) losses due to past changes of rate and seasonality in accumulation at Dome C, which are both not well constrained at Dome C. The large variability of the raw and smoothed [I] record (Fig.1) likely is a reflection of inter-annual variability of accumulation at the site. Note that even if annual accumulation remains unchanged a shift of snow fall from summer/winter to spring, the main ozone hole season, would have profound impacts on the magnitude of the photochemical loss of iodine. In the paper this aspect remains unexplored.

ii) more importantly though, Legrand et al. (2018) concluded that volatilization of [I] must have taken place either during storage or melting of firn during analysis and therefore discuss in their work only ice concentrations. The 13.72m shallow core from Dome C used in this study, well above the firn-ice transition at 80m, consists entirely of firn and snow, where the 3m top section containing the [I] trend discussed has also the lowest densities, thus is most prone to such losses. While no significant iodine loss during the analytical run (≈ 12 hours) was reported I did not find any details in the paper or SM regarding transport and storage conditions, and in particular the time between core and pit sample collection and analysis.

One could of course assume that all core and pit samples were subject to similar loss rates, and relative [I] changes would have remained the same. However, comparison of core and pit records raise some suspicion that samples near the top may have been affected by such loss. E.g. Fig.S3 bottom panel - the negative [I] trend in Sp2017 is largely driven by a distinct decrease in [I] variability above 1m snow depth (\sim past 10yr), with (summer?) [I] minima remaining about the same compared to the entire pit record, whereas winter time [I] maxima are much reduced, this is puzzling because this change started only 10yr prior i.e. 2007 (1m), unrelated to TOC. Fig.S4 - Sp2017 shows significantly larger [I] variability compared to IC2012 during the period of overlap. And SP2013 is entirely flat near the top. These differences are not critically discussed, but may reflect loss during transport/storage or possibly spatial variability. Indeed, considering only the Sp2017 profile one would place the inflection point at 1m depth (2007), unrelated to any concurrent TOC change. Thus, was the timing in between sample collection and analysis different comparing IC2012, SP2017 and Sp2012?

Some conclusions/ implications are in my view overstated. The authors suggest that that iodine-related loss of tropospheric ozone is more active now than pre-ozone hole (L321-323), however recent observations indicate net production of tropospheric ozone and not loss above EAIS (e.g. Legrand et al., 2016), and increases of summer tropospheric ozone mixing ratios above EAIS compared to the pre-1975 period are linked to the modern ozone hole (Jones & Wolff, 2003); so if iodine induced ozone loss plays a role it is superseded by far by the increase of the snowpack NO_x source and subsequent increase of atmospheric oxidising capacity.

The statement of significant redistribution of iodine and transport to the coast via katabatic outflow is not supported by the cited study (Schoenhardt et al., 2012), which reports satellite derived vertical IO columns above Antarctica. Fig.1 in Schoenhardt et al (2012) shows IO (2004-09 means) mostly above sea ice, coast lines, ice shelves (Ross and Weddell sea), and parts of the continent between South Pole and Dronning Maud Land / Weddell Sea sector, but no signal in the sector of Terre Adélie, in particular none between the study site Dome C and the coast, which has one of the strongest katabatic outflows of the continent.

In summary some of the reported [I] values, especially in the upper section most relevant for the presented interpretation, may have been subject to loss between sample collection and analysis. This needs to be clarified before going any further. Therefore the claim that ice core iodine may be suitable as a proxy of past stratospheric ozone trends is at this stage premature.

REFERENCES

Grannas et al., 2007: An overview of snow photochemistry: evidence, mechanisms and impacts, *Atmos. Chem. Phys.*, 7(16), pp.4329--4373, doi:10.5194/acp-7-4329-2007, 2007.

Jones, A.E. and Wolff, E.W.: An analysis of the oxidation potential of the South Pole boundary layer and the influence of stratospheric ozone depletion, *J. Geophys. Res.*, 108(D18), 2003.

Legrand et al.: Alpine ice evidence of a three-fold increase in atmospheric iodine deposition since 1950 in Europe due to increasing oceanic emissions, *Proc. Natl. Acad. Sci.*, 115(48), pp.12136-12141, doi:10.1073/pnas.1809867115, 2018.

Legrand et al.: Inter-annual variability of surface ozone at coastal (Dumont d'Urville, 2004-2014) and inland (Concordia, 2007-2014) sites in East Antarctica, *Atmos. Chem. Phys.*, 16(12), pp.8053--8069, doi:10.5194/acp-16-8053-2016, 2016.

Mayewski, P.A. and Legrand, M.: Recent Increase in Nitrate Concentration of Antarctic Snow, 346, pp.258-260, *Nature*, 1990.

Savarino et al.: Nitrogen and oxygen isotopic constraints on the origin of atmospheric nitrate in coastal Antarctica, *Atmos. Chem. Phys.*, 7(8), 1925-1945, 2007.

Response to reviewer's comments

We are grateful for the feedback of the three reviewers and the guidance of the Editor, Dr. Bolles. Their comments and concerns have been addressed robustly, as detailed below. Our responses (i.e. changes to the manuscript and information) follow each reviewer comment in red.

REVIEWER #1 (REMARKS TO THE AUTHOR):

In this study, an iodine decrease in a shallow ice core from Dome C is used as a proxy for ozone hole depletion in the latter half of the 20th century. Laboratory measurements, atmospheric modelling, and comparison to the ice core sodium record are used to validate this interpretation. The authors provide a clear and concise explanation of their findings. This study should be published because it is the first time, to my knowledge, that the reemission of iodine from surface snow has been convincingly used as a proxy for increased UV-radiation reaching the Antarctic ice sheet surface. According to the authors, this is also only the third Antarctic iodine record to be published, and these other two records were incapable of reconstructing ozone hole depletion. The science described in this manuscript is sound. This interpretation of iodine could be very useful for future ice core studies. I just have a few minor comments. My only major comment is that the writing in both the main text and the supplement should be somewhat improved prior to publication. Some examples of possible grammatical changes are listed at the end of this review.

We thank the reviewer for his/her positive comments. The manuscript has been edited by professional English editors.

Major comments

The authors explain that UV-radiation reaching the snow surface does not affect the sodium record. Therefore, the sodium record is a good component to compare to the iodine record because otherwise, the temporal variability two records should (and do) correlate quite closely. Are there other halogens that are affected by the ozone hole that could or have been measured in ice cores?

We focused on iodine since “*Under laboratory and simulated Antarctic radiation conditions, it was observed that the iodate UV-absorption, and its subsequent photoreduction ($IO_3^-_{(aq)} + hv \rightarrow IO_{(g)} + O_2^-_{(aq)}$), was enhanced in the 275-400 nm range, with a maximum peak at 295 nm 31....Similar conclusions were reported also for the photooxidation of iodide into $I_{2(g)}$ in real snow samples spiked with known amounts of iodide and exposed to Antarctic sunlight ($O_{2(aq)} + 4H^+_{(aq)} + 6I^-_{(aq)} \rightarrow 2H_2O + 2I_3^-_{(aq)}$; $I_3^-_{(aq)} \leftrightarrow I_{2(g)} + I^-_{(aq)}$...encompassing the same spectral range where the largest changes in incoming UV-radiation were observed between the pre- and post-ozone hole periods”.* Other halogens such as Cl and Br could potentially also be influenced by the change in the ozone hole however their

absorption spectra are shifted to the deep UV (<250 nm) much more than the iodine one. Therefore, their UV absorption is outside the *spectral range where the largest changes in incoming UV-radiation were observed between the pre- and post-ozone hole periods (280-330 nm)*.

In the supplement, the authors emphasize that this is only the third Antarctic iodine record to be published. It is also the only one that can be used to reconstruct ozone hole depletion. This is an excellent point. Can it be emphasized in the introduction somehow?

Thanks. We added a new sentence to emphasize it in the Introduction (line 64-66)

Minor comments

We really thank the reviewer for the careful reading of the paper. All minor changes and suggestions below have now been implemented in the revised manuscript. In addition, professional English editors have revised the entire manuscript.

Change “well-studied” to “well-established” in the introduction and abstract

Line 69: remove “at present”

Lines 84-85: “due to its very dry..” and “its very thin...”

Line 92: “degradation during transport or after deposition”

Lines 93-95: “To support to the 13.72 m-deep shallow core, a 1.3 m-deep snow pit (Sp2013) and a 4 m-deep (Sp2017) snow pit were dug...”

Line 140: “mostly in ionic form as iodide...”

Lines 97-98: “we used the Change-Point Analysis test”

Line 199-200: change to “it was” in all cases

Line 201: change to “worst”

Line 203: change to “but will not affect the”

Line 216-220: confusing, please clarify

Line 223: “If a relationship between the grain size and iodine concentration existed,”

Line 225: “but we observe”

Line 228: “on timescales”

Line 229: Continuous measurements of what?

Line 231: “at the surface”

Line 233: “is detected”

Line 242: “2-3 years of burial”

Line 248: “Evidence that”

Line 249: “also comes from”

Line 259: “to 3.5 m”

Line 272: “last few decades”

Lines 286-287: “All of these previous studies”

Lines 298-299: “mid-latitude oceans, and significantly decreasing emissions towards”

Lines 299-300: please clarify: “In the polar regions, iodine sources mainly and first year sea ice”

Line 311: “has altered”

Lines 236-239: “Radiative transfer within the snowpack will be further discussed in Section 4.2.3, but its rapidity suggests that snow metamorphism, its physical characteristics and sintering process of the snow pack play a negligible role in describing the observed iodine trend.” What is “its” referring to here? Please clarify this sentence.

REVIEWER #2 (REMARKS TO THE AUTHOR):

In the manuscript, the authors describe results of iodine determinations in ice cores from the Dome C station, which is located on the polar plateau of East Antarctica at about 3000 m altitude. The core discussed covers a little more than 200 years. The main result of the measurements is the observed decrease in total iodine concentrations in the ice since about the mid-1970s, which the authors link to the occurrence of the Antarctic ozone hole and the associated increase in UV radiation in this region. The authors' central hypothesis is that the increased UV intensity drives photochemical processes in the upper ice layers that accelerate the release of the halogen into the gas phase, thereby interfering with the geochemical cycles of the trace element. The hypothesis is supported by the discussion presented in various chapters of parameters such as UV intensity reaching the ground, photochemistry of various iodine compounds in the condensed phase, change in snow accumulation or physical characteristics of snow, transport mechanisms, etc.

Without question, iodine is an interesting element which not only shows an active redox chemistry, but the involved reduced and oxidized species also show different volatility and thus can migrate between different spheres (biosphere, pedosphere, hydrosphere, atmosphere, cryosphere), i.e. the element has the prerequisites to possess an active biogeochemical cycle. In fact, iodine has already been proposed as a paleoredox indicator. In my opinion, the paper is an excellent scientific work, in which, based on a comprehensive data set using various tools (trace analytical methods, chemical and transport models), an interesting relationship is demonstrated, in this case on the complex influence of photochemically-induced processes on important biogeochemical trace substance cycles. Therefore, I would recommend publishing the manuscript after minor changes.

We thank the reviewer for his/her positive comments.

Page 9: line 285: “The decoupling observed since 1975 is inconsistent with an iodine decrease driven by changes in atmospheric transport.”!?

Page 10, line 299: The wording needs to be changed here, the sentence makes no sense as it is.

Perhaps: “In the polar regions, iodine sources are mainly connected to first year sea ice, with both macroalgae and microalgae living under the sea ice and abiotic activation of iodine on sea ice and snow surfaces playing a role”!?

Thanks. We modified the above two sentences according to reviewer’s suggestions. In addition, professional English editors have revised the entire manuscript.

Page 10, line 320: While tropospheric ozone depletion is well supported with citations (which is only of regional importance), the influence of iodine on atmospheric particle formation (which may also have supraregional importance) is only very sparsely supported and cited. Here one could still improve.

Thank you for noting this. We agree, and we have now added some relevant citations (below) to support the role of iodine in atmospheric particle formation.

O'Dowd, C.D., et al., *Marine aerosol formation from biogenic iodine emissions*. Nature, 2002. 417(6889): p. 632-636.

Sipilä, M., et al., *Molecular-scale evidence of aerosol particle formation via sequential addition of HIO₃*. Nature, 2016. 537: p. 532.

Baccarini, A., et al., *Frequent new particle formation over the high Arctic pack ice by enhanced iodine emissions*. Nature Communications, 2020. 11(1): p. 4924.

REVIEWER #3 (REMARKS TO THE AUTHOR):

This study reports the first iodine firn core record from the East Antarctic Ice Sheet (EAIS) that covers about the last 200 years. The results indicate that the firn core iodine concentrations remained fairly constant during the pre-ozone hole period (1800-1974 CE) but have declined twofold since the onset of the ozone hole era (~1975 CE), closely tracking the stratospheric ozone above Antarctica. This is an interesting study highlighting the potential impact on iodine chemistry above Antarctica due to enhanced surface-UV from the modern ozone hole. The chosen statistical methods and modelling approach are sound and yield convincing results. My concerns are related to the presented firn core concentrations of [I].

We thank the reviewer for appreciating the manuscript and the methodological approach used in the study. We fully considered the concerns rose from the reviewer regarding the iodine signal preservation

in the firn core and we added a new detailed section (S7) to the Supplementary Information addressing this point.

It is well known that post-depositional exchange driven by photochemical and physical processes alters concentrations of a range of snow impurities before archival in firn and ice, and this has been the subject of numerous studies (e.g. Grannas et al., 2007). A quantitative understanding of the air-snow exchange is required before attempting to interpret the ice core record in terms of past atmospheric change.

Indeed, the aim of the paper is to understand the air-snow exchange and its possible changes during the ozone hole period. All the known possible processes that could affect the iodine emission rate have been taken into account and discussed in the manuscript. In particular, in section 4.2.1 we discussed the “Change in snow accumulation”, in section 4.2.2 the “Role of snow pack physical characteristics”, in 4.2.3 the “Inner snow pack post-depositional processes and UV light penetration” in 4.2.4 the “Changes in the transport mechanism from the coast to the site” and in 4.2.5 the “Possible changes in the strength of coastal source emissions”. Furthermore, the Supplementary Material includes a comparison with other iodine Antarctic records (S4. Comparison with other iodine ice core records). Regarding the reviewer’s concerns about the sample storage, we have now described in the Supplementary how the samples for the three records were treated after being collected and we have also discussed the possible impact of sample transportation and storage on the preservation of the iodine signal (S7. Possible effect of ice core sample transportation and snow sample processing on iodine loss).

Not doing so can lead to premature interpretation, as illustrated by the case of nitrate, where the modern increase in nitrate seen in EAIS pit profiles was linked to polar stratospheric clouds and the ozone hole, whereas later research showed that nitrate spikes in the low-accumulation regions reflect post-depositional air-snow recycling rather than change in an atmospheric source (e.g. Savarino et al., 2007).

We understand the point of the reviewer, but note that nitrogen and iodine species have rather different chemical cycles in the atmosphere and in the surface snow pack. Iodine is removed from the snow pack in a few hours after deposition as a consequence of the solar radiation reaching the snow surface (see Spolaor et al. 2014, 2017 and 2019). During the polar night, iodine is preserved in the snowpack. A clear example of signal preservation can be seen in the Law Dome ice core where the iodine signal shows a clear seasonal pattern in both firn and ice (Vallelonga et al 2017; Spolaor et al 2014). We already introduced this concept in the original manuscript at lines 262-266. When compared to nitrate, iodine exhibits a different profile (see Figure S4 in the supplementary material). Here, iodine decreases, suggesting a direct emission into the atmosphere from the snow pack (i.e. due to the enhanced solar UV-

radiation caused by the ozone hole), while nitrate increases suggesting a recycling in the upper snow pack. Considering that the aim of this manuscript is to explore the causes that drive the iodine decrease since 1975, we did not focus on the nitrate chemistry.

Spolaor, A., et al., *Seasonality of halogen deposition in polar snow and ice*. Atmos. Chem. Phys., 2014. **14**: p. 9613-9622.

Spolaor, A., et al., *Feedback mechanisms between snow and atmospheric mercury: Results and observations from field campaigns on the Antarctic plateau*. Chemosphere, 2018. **197**: p.

Spolaor, A., et al., *Diurnal cycle of iodine, bromine, and mercury concentrations in Svalbard surface snow*. Atmos. Chem. Phys., 2019. **19**(20): p. 13325-13339.

Vallelonga, P., et al., *Sea-ice-related halogen enrichment at Law Dome, coastal East Antarctica*. Clim. Past, 2017. **13**(2): p. 171-184.

Total iodine in surface snow is in so far complicated because it is subject to both photochemical loss as well as physical volatilization. In particular Legrand et al. (2018) found that due to the high volatility of some of the iodine species such as hypoiodous acid (HOI), iodine concentrations in ice strongly depend on the annual snowfall rate, particularly at sites with low snowfall rates.

We know that the accumulation rate can affect the photochemical iodine loss and, as already pointed out in the original main text (lines 196 - 204), we carefully evaluated this point showing that the change in snow accumulation have not exceeded 10% over the last 200 years. As reported in section 4.2.1 of the original main text: “*the average snow accumulation rate computed during different studies are very similar: for the period 1955–1998 it was 26 ± 1.3 mm w.e. yr⁻¹, for the period 1816-1998 it was 25 ± 1.3 mm w.e. yr⁻¹ 35 and for the period 2006-2013³⁴ it was 27 mm w.e. yr⁻¹. This leads to the conclusion that the snow accumulation rate at this location has not changed significantly over the last 200 years, and that in the worst scenario it might have affected the iodine signal for no more than 10%*”. Therefore, even in the worst scenario, this change of not more than 10% cannot explain the twofold decline in I concentration that we detected in the firn record since the onset of the ozone hole era.

Thus, I am concerned that the presented results are affected by such post-depositional loss in two ways:

i) losses due to past changes of rate and seasonality in accumulation at Dome C, which are both not well constrained at Dome C. The large variability of the raw and smoothed [I] record (Fig.1) likely

is a reflection of inter-annual variability of accumulation at the site. Note that even if annual accumulation remains unchanged a shift of snow fall from summer/winter to spring, the main ozone hole season, would have profound impacts on the magnitude of the photochemical loss of iodine. In the paper this aspect remains unexplored.

As stated above and according to all the available accumulation data (more than 20 cores with measurements of b-radioactive reference horizons of January 1955 and January 1965, and Tambora 1815 eruption marker, 4 networks of more than 40 stakes each around Dome C, one since 1996 and other 3 networks since 2005), the average accumulation at Dome C has not changed more than 10% over the last 200 years (Frezzotti et al., 2005; Urbini et al., 2008; Genthon et al., 2016; GLACIOCLIM-SAMBA accumulation stakes).

Regarding the seasonality (added at line 209-219), combining a Lagrangian trajectory model output and the European Re-Analysis ERA40 data (from 1980 to 2011), it is reported that approximately 35-40% of the annual precipitation at Dome C occurred during the summer season (December to February) with the remaining 60-65% being homogenously distributed among the remaining months (Scarchilli et al., 2011).

One may argue that the seasonal precipitation pattern might change on a yearly basis. However, we need to consider that:

- a) UV can penetrate the snowpack for the first 20-30 cm (as stated in the original manuscript, section 4.2.3), which corresponds to 2-3 years of burial snow due to the low snow accumulation rate at Dome C (Picard et al 2019, France et al 2011 and GLACIOCLIM-SAMBA stake network, <http://pp.ige-grenoble.fr/pageperso/faviervi/dc.php> website). Independently of the seasonal precipitation pattern, all the iodine present in the annual snow layers would experience a similar photochemical activation, thereby greatly minimizing the hypothetical role of a seasonal precipitation pattern change.
- b) A recent study by Picard et al., 2019 highlights that “the age distribution of snow on the surface spans over more than a year” because of wind redistribution. The wind redistribution effect does not affect the long-term iodine trend determined at a decadal scale, as highlighted in the text at line 213-216, and through the comparison of the snow pits and the shallow ice core (Table S2). Besides, note that the effect of wind redistribution has already been taken into account when obtaining the ice core age scale (Frezzotti 2005, Gautier et 2016).

Overall, these evidences show that the seasonal precipitation pattern cannot explain the distinct gradual decrease, by a factor of 2, in iodine concentrations since the onset of the Antarctic ozone hole in the mid-70s. We added the discussion about the precipitation seasonality in section 4.2.1 (main text).

Scarchilli, C., M. Frezzotti, and P. Ruti, *Snow precipitation at four ice core sites in East Antarctica: provenance, seasonality and blocking factors*. *Climate Dynamics*, 2011. **37**(9): p. 2107-2125.

Picard, G., et al., *Observation of the process of snow accumulation on the Antarctic Plateau by time lapse laser scanning*. *The Cryosphere*, 2019. **13**(7): p. 1983-1999.

Gautier, E., et al., *Variability of sulfate signal in ice core records based on five replicate cores*. *Clim. Past*, 2016. **12**(1): p. 103-113.

Frezzotti M., Pourchet M., Flora O., Gandolfi S., Gay M., Urbini S., Vincent C., Becagli S., Gagnani R., Proposito M., Severi M., Traversi R., Udisti R., Fily M. (2005) *Spatial and temporal variability of snow accumulation in East Antarctica from traverse data*. *J. Glaciol.*, 51(172), 113-124, doi: 10.3189/172756505781829502

France, J.L., et al., *Snow optical properties at Dome C (Concordia), Antarctica; implications for snow emissions and snow chemistry of reactive nitrogen*. *Atmos. Chem. Phys.*, 2011. **11**(18): p. 9787-9801.

Genthon, C., et al., *Meteorological and snow accumulation gradients across Dome C, East Antarctic plateau*. *International Journal of Climatology*, 2016. **36**(1): p. 455-466.

Urbini, S., et al., *Historical behaviour of Dome C and Talos Dome (East Antarctica) as investigated by snow accumulation and ice velocity measurements*. *Global and Planetary Change*, 2008. **60**(3): p. 576-588.

ii) more importantly though, Legrand et al. (2018) concluded that volatilization of [I] must have taken place either during storage or melting of firn during analysis and therefore discuss in their work only ice concentrations. The 13.72m shallow core from Dome C used in this study, well above the firn-ice transition at 80m, consists entirely of firn and snow, where the 3m top section containing the [I] trend discussed has also the lowest densities, thus is most prone to such losses. While no significant iodine loss during the analytical run (≈ 12 hours) was reported I did not find any details in the paper or SM regarding transport and storage conditions, and in particular the time between core and pit sample collection and analysis.

Thanks for highlighting this important point, which we had already thoroughly considered during the sample collection and analysis procedure of the DOME-C record, as it is standard practice in the ice core handling and processing in our field. We have now added new materials to the revised manuscript to

give more details on how the samples were collected and treated before and during the analysis. These details are now included in a new section (S7) in the Supplementary Information:

- a) Considering that iodine can be easily released from ice/firn samples directly exposed to external radiation (see Spolaor et al., 2013, Figure 9), the samples were carefully stored under dark conditions until the analysis. To better clarify how samples were treated, we added an additional section in the Supplementary Information (see S7). To support the ice core and firn capability to preserve the iodine signal, we refer first to the Law Dome record where the seasonality of iodine (see Vallelonga et al. 2017 and Spolaor et al., 2014) is preserved in both firn and ice samples, suggesting that the preservation of the signal is obtained if the samples are kept in dark conditions and frozen until the analysis. Besides, iodine signal preservation in low-density samples (surface snow) was also demonstrated from our previous snow sampling studies that detected iodine day-to-night variability (Fig. 3g in Spolaor et al., 2019). The time encompassed between sample collection and analysis was similar to that in the Dome C study. The clear difference in iodine signal determined between the day and night surface snow samples in our previous work using the same methodology (Spolaor et al., 2019) strongly supports our approach at DOME C, therefore demonstrating that the method adopted for transportation and storage before the analysis preserves the iodine signal in the snow\firn\ice.

- b) The reviewer suggests that the iodine decrease in the upper 3.5 m is due to the lower density of the firn. The density in the upper part of the core (0-2 m) ranges between 0.32 to 0.35 kgL⁻¹ increasing from 0.35 to 0.38 kgL⁻¹ at 3.5 m depth, when the divergent trend between iodine and sodium was recorded (divergence determined from an independent and robust statistical test). The density in the deepest part of the shallow core ranges between 0.45 to 0.48 kgL⁻¹. This slight change in density in the upper part ($\approx 15\%$) cannot justify the observed two-fold decline in iodine in the topmost 3.5 meters neither during transport nor storage. In addition, looking at Fig. S2 from Legrand et al., 2018, a density-driven decrease in iodine concentration was observed from density values higher (almost double) than the ones that we have in our record, i.e. at the firn-ice transition (density = 0.83 KgL⁻¹). However, we did not observe this decrease at the firn-ice transition, but within the firn itself. Also, the firn-ice transition at Dome C is at 80 m (Leduc-Leballeur et al., 2015, Fig 1).

In summary, the analytical procedure that we followed during sample collection, storage, processing and analysis was standardized and successfully applied in many previous studies that also involved iodine quantification from firn and snow matrices (e.g. Vallelonga et al., 2017; Spolaor et al., 2019, Cuevas et al., 2018, Spolaor et al., 2013, Corella et al., 2019). For this reason, we exclude that iodine volatilization during storage or melting of firn during analysis can justify the observed trend because:

- a) iodine concentration was monitored during the analytical run through the multiple analysis of selected samples and no significant iodine losses were observed during the analysis (12 hours);
- b) if dark conditions are guaranteed during storage and transport (as it was for the core and snow pit samples of this study) iodine losses are minimized. Indeed, iodine losses from snow/ice are favoured when samples are exposed to light (e.g. Spolaor et al., 2019, Kim et al., 2016 and Spolaor et al., 2013). Anyway, if they had happened, they would have affected the whole core in a similar way due to the slight density differences recorded and not only in the upper 3.5 m.
- c) Lastly, if density had driven the decrease in iodine concentration, we would have expected an iodine-decreasing trend along the entire record and not only in the uppermost meters.

Spolaor, A., et al., *Seasonality of halogen deposition in polar snow and ice*. Atmos. Chem. Phys., 2014. **14**: p. 9613-9622.

Corella, J.P., et al., *Holocene atmospheric iodine evolution over the North Atlantic*. Clim. Past, 2019. **15**(6): p. 2019-2030.

Vallelonga, P., et al., *Sea-ice-related halogen enrichment at Law Dome, coastal East Antarctica*. Clim. Past, 2017. **13**(2): p. 171-184.

M. Leduc-Leballeur *et al.*, "Modeling L-Band Brightness Temperature at Dome C in Antarctica and Comparison With SMOS Observations," in *IEEE Transactions on Geoscience and Remote Sensing*, vol. 53, no. 7, pp. 4022-4032, July 2015, doi: 10.1109/TGRS.2015.2388790.

Spolaor A, Barbaro E, Cappelletti D, Turetta C, Mazzola M, Giardi F, Björkman MP, Lucchetta F, Dallo F, Pfaffhuber KA, Angot H. Diurnal cycle of iodine, bromine, and mercury concentrations in Svalbard surface snow. *Atmospheric Chemistry and Physics*. 2019 Oct 29;19(20):13325-39.

Picard, G., et al., *Observation of the process of snow accumulation on the Antarctic Plateau by time lapse laser scanning*. The Cryosphere, 2019. **13**(7): p. 1983-1999.

Kim K, Yabushita A, Okumura M, Saiz-Lopez A, Cuevas CA, Blaszcak-Boxe CS, Min DW, Yoon HI, Choi W. Production of molecular iodine and tri-iodide in the frozen solution of iodide: implication for polar atmosphere. *Environmental science & technology*. 2016 Feb 2;50(3):1280-7.

Cuevas, C.A., et al., *Rapid increase in atmospheric iodine levels in the North Atlantic since the mid-20th century*. Nature Communications, 2018. **9**(1): p. 1452.

Spolaor, A., et al., *Sea ice dynamics influence halogen deposition to Svalbard*. The Cryosphere, 2013. 7(5): p. 1645-1658.

One could of course assume that all core and pit samples were subject to similar loss rates, and relative [I] changes would have remained the same. However, comparison of core and pit records raise some suspicion that samples near the top may have been affected by such loss. E.g. Fig.S3 bottom panel - the negative [I] trend in Sp2017 is largely driven by a distinct decrease in [I] variability above 1m snow depth (~ past 10yr), with (summer?) [I] minima remaining about the same compared to the entire pit record, whereas winter time [I] maxima are much reduced, this is puzzling because this change started only 10yr prior i.e. 2007 (1m), unrelated to TOC.

It is not possible to discern summer minima and winter maxima from any snow pit or ice core record collected at Dome C due to the low accumulation and the snow redistribution at the surface (see Picard et al., 2019 and our previous answer regarding the snow accumulation). What we observed is the result of one year (or more) of surface precipitation redistributed by wind scouring. The differences between the ice core and the snow pits samples are due both to different sampling resolution (higher than in the firm core, see Methods section in the main text and S2) and spatial variability. The comparison between the ice core and the snow pits, was performed to validate the robustness of the results obtained from the ice core record during the ozone-hole period. The similarities in concentration retrieved from the three records (see Table S2) reinforce our hypothesis of ozone-hole driven decrease in surface iodine since 1975. Lastly, considering the time elapsed between sample collection, storage (see our answer to storage conditions) and analysis for the three different records, we exclude that what we observe is affected by iodine losses during storage. More details about this are given in the supplementary (S7) and in the following answer.

Picard, G., et al., *Observation of the process of snow accumulation on the Antarctic Plateau by time lapse laser scanning*. The Cryosphere, 2019. 13(7): p. 1983-1999.

Fig.S4 - Sp2017 shows significantly larger [I] variability compared to IC2012 during the period of overlap. And SP2013 is entirely flat near the top. These differences are not critically discussed, but may reflect loss during transport/storage or possibly spatial variability. Indeed, considering only the Sp2017 profile one would place the inflection point at 1m depth (2007), unrelated to any concurrent TOC change. Thus, was the timing in between sample collection and analysis different comparing IC2012, SP2017 and Sp2012?

The differences among Sp2013, IC2012 and Sp2017 most likely reflect spatial variability and the difference sampling resolution than losses during transport/storage (as explained in our previous

answers). For this reason, the three records (for both iodine and sodium) were merged as shown in Figure 3 of the original manuscript. The Sp2013 and Sp2017 were included in this study to evaluate whether the average iodine concentrations (Table S1) were similar and consistent with the ones found in the shallow core. We cannot use the results obtained from the Sp2017 for the tipping-point detection since the record covers only the ozone-hole period. Anyway, we understand the concerns of the reviewer regarding the possible iodine loss from the collection to the analysis. For this reason, and as detailed in our previous responses, we added a separate chapter in the Supplementary Information to further clarify this point and to provide more details about the timing between sample collection and analysis (see S7). In details, the firm core samples were analysed 32 months after collection, Sp2013 samples were analysed 13 months after collection and Sp2017 samples were analysed 14 months after collection. The lowest iodine concentrations were recorded in the Sp2013 samples ($0.026 \pm 0.020 \text{ ng g}^{-1}$, see Table S2 for additional details), where the timing between sampling and the analysis was the shortest compared to the other records. This suggests that the time elapsed between the sampling and the analysis does not seem to play a role in affecting the iodine concentration and that the flat profile observed in Sp2013 is due to spatial variability rather than sample loss (otherwise we would have expected a similar behaviour in the other records that were analysed a longer time after collection than Sp2013, which was not the case). Therefore, the inter-annual time scale differences among the cores are only due to inter-annual variability and they do not reflect loss during transport/storage.

Some conclusions/ implications are in my view overstated. The authors suggest that that iodine-related loss of tropospheric ozone is more active now than pre-ozone hole (L321-323), however recent observations indicate net production of tropospheric ozone and not loss above EAIS (e.g. Legrand et al., 2016), and increases of summer tropospheric ozone mixing ratios above EAIS compared to the pre-1975 period are linked to the modern ozone hole (Jones & Wolff, 2003); so if iodine induced ozone loss plays a role it is superseded by far by the increase of the snowpack NOx source and subsequent increase of atmospheric oxidising capacity.

We respectfully disagree with the reviewer because there is not overstatement here. We simply say that more atmospheric iodine, as a result of more reemission of iodine from surface snow, leads to more “iodine-related tropospheric ozone loss over Antarctica”, since iodine is a very efficient ozone depleting substance and its effect on ozone loss would linearly depend on the levels of atmospheric iodine. Therefore, our statement is literally correct. We do not speak about net ozone production or loss, as inferred by the reviewer and which is out of the scope of the manuscript, but on iodine-related ozone loss. Nevertheless, and to avoid confusion, we have removed this sentence in the revised manuscript.

The statement of significant redistribution of iodine and transport to the coast via katabatic outflow is not supported by the cited study (Schoenhardt et al., 2012), which reports satellite

derived vertical IO columns above Antarctica. Fig.1 in Schonehardt et al (2012) shows IO (2004-09 means) mostly above sea ice, coast lines, ice shelves (Ross and Weddell sea), and parts of the continent between South Pole and Dronning Maud Land / Weddell Sea sector, but no signal in the sector of Terre Adélie, in particular none between the study site Dome C and the coast, which has one of the strongest katabatic outflows of the continent.

Schonehardt et al (2012) remarkably show that reactive atmospheric iodine is ubiquitous and widespread over Antarctica, for which there is currently no mechanistic explanation. The purpose of this statement (line 321 - 325 of the original manuscript) was to highlight one potential implication of our study which is that the surface reemission of iodine from central Antarctica and subsequent redistribution by atmospheric transport could potentially be a source of the widespread distribution of iodine observed by the satellite measurements of Schonehardt et al (2012) over the entire Antarctic continent. This sentence has now been rephrased (333-336) for clarity in the revised manuscript.

In summary some of the reported [I] values, especially in the upper section most relevant for the presented interpretation, may have been subject to loss between sample collection and analysis. This needs to be clarified before going any further. Therefore the claim that ice core iodine may be suitable as a proxy of past stratospheric ozone trends is at this stage premature.

We addressed all the concerns raised by the reviewer related to the possible iodine loss during storage and we added a dedicated section in the Supplementary (see S7) where the possible effect of sample storage on iodine loss is discussed and ruled out. Based on our experience and well-proven methodology for ice core sampling and iodine analysis from ice cores, as explained in our responses above and as documented by our many previous publications on the subject, we exclude that the observed two-fold decline in iodine concentration can be explained by iodine losses during transport and storage. On these grounds, at the best of our knowledge and after excluding all the known competing processes that could have explained the recorded iodine behaviour, we support the hypothesis originally presented in the main text, i.e. *the enhanced incident UV radiation due to the ozone hole has caused a continuous decline in iodine concentration in ice in inner Antarctica since ~1975 and that given the direct link observed between ice core iodine at Dome C and stratospheric ozone hole evolution [...] the ice core iodine present on the Antarctic Plateau may potentially serve as an archive for past stratospheric ozone changes at centennial to millennial time scales.*

REFERENCES

Grannas et al., 2007: An overview of snow photochemistry: evidence, mechanisms and impacts, Atmos. Chem. Phys., 7(16), pp.4329--4373, doi:10.5194/acp-7-4329-2007, 2007.

Jones, A.E. and Wolff, E.W.: An analysis of the oxidation potential of the South Pole boundary layer and the influence of stratospheric ozone depletion, *J. Geophys. Res.*, 108(D18), 2003.

Legrand et al.: Alpine ice evidence of a three-fold increase in atmospheric iodine deposition since 1950 in Europe due to increasing oceanic emissions, *Proc. Natl. Acad. Sci.*, 115(48), pp.12136-12141, doi:10.1073/pnas.1809867115, 2018.

Legrand et al.: Inter-annual variability of surface ozone at coastal (Dumont d'Urville, 2004-2014) and inland (Concordia, 2007-2014) sites in East Antarctica, *Atmos. Chem. Phys.*, 16(12), pp.8053--8069, doi:10.5194/acp-16-8053-2016, 2016.

Mayewski, P.A. and Legrand, M.: Recent Increase in Nitrate Concentration of Antarctic Snow, 346, pp.258-260, *Nature*, 1990.

Savarino et al.: Nitrogen and oxygen isotopic constraints on the origin of atmospheric nitrate in coastal Antarctica, *Atmos. Chem. Phys.*, 7(8), 1925-1945, 2007.

Reviewers' Comments:

Reviewer #3:

Remarks to the Author:

The authors are commended for addressing my comments in a very comprehensive manner and clarifying the manuscript, in particular regarding potential [I] changes during transport and storage. I support publication of this very interesting study with a few minor suggestions below:

- L209-219 Thanks for clarifying. A comparison of the time scales of local burial rate and [I] loss rate from sun lit surface snow based on the author's previous work would be useful here. In the absence of a sensitivity study with an air-snow model the potential impact of systematic changes in precipitation seasonality over several years remains an uncertainty for the interpretation of future [I] core records, likely contributing to the observed spatial variability (firn core, various snow pits), and this should be noted.

- The main driver of [I] loss from snow/firn appears to be radiation, whereas large temperature changes/ gradients may have also an impact. It is straight forward to keep an ice sample in the dark, but less so at constant temperature. It therefore would be useful to add a sentence to S7 regarding potential impacts of temperature changes during storage (supposedly always below -20°C) based on the author's experience/ previous work.

Response to referees

- L209-219 Thanks for clarifying. A comparison of the time scales of local burial rate and [I] loss rate from sun lit surface snow based on the author's previous work would be useful here. In the absence of a sensitivity study with an air-snow model the potential impact of systematic changes in precipitation seasonality over several years remains an uncertainty for the interpretation of future [I] core records, likely contributing to the observed spatial variability (firn core, various snow pits), and this should be noted.

We added this sentence at L219: "However, the potential impact of systematic changes in precipitation seasonality over several years remains an uncertainty for the interpretation of future iodine ice core records and might contribute to the observed spatial variability".

- The main driver of [I] loss from snow/firn appears to be radiation, whereas large temperature changes/ gradients may have also an impact. It is straightforward to keep an ice sample in the dark, but less so at constant temperature. It therefore would be useful to add a sentence to S7 regarding potential impacts of temperature changes during storage (supposedly always below -20°C) based on the author's experience/ previous work.

We added this sentence in section S7 (line 311-314) as suggested by the reviewer: "We underline that a correct sample storage is fundamental to minimize iodine losses from ice core and snow samples. To assure reliable sample representativeness, the cold chain (temperature equal or below -20°C) as well as dark conditions must be guaranteed for the entire transportation process".